# On Mixup Training: Improved Calibration and Predictive Uncertainty for Deep Neural Networks

**Sunil Thulasidasan**[*,1,2], **Gopinath Chennupati**[1], **Jeff Bilmes**[2],
**Tanmoy Bhattacharya**[1], **Sarah Michalak**[1]

[1]Los Alamos National Laboratory
[2]Department of Electrical and Computer Engineering, University of Washington

## Abstract

Mixup [40] is a recently proposed method for training deep neural networks where additional samples are generated during training by convexly combining random pairs of images and their associated labels. While simple to implement, it has been shown to be a surprisingly effective method of data augmentation for image classification: DNNs trained with mixup show noticeable gains in classification performance on a number of image classification benchmarks. In this work, we discuss a hitherto untouched aspect of mixup training – the calibration and predictive uncertainty of models trained with mixup. We find that DNNs trained with mixup are significantly better calibrated – i.e., the predicted softmax scores are much better indicators of the actual likelihood of a correct prediction – than DNNs trained in the regular fashion. We conduct experiments on a number of image classification architectures and datasets – including large-scale datasets like ImageNet – and find this to be the case. Additionally, we find that merely mixing features does not result in the same calibration benefit and that the label smoothing in mixup training plays a significant role in improving calibration. Finally, we also observe that mixup-trained DNNs are less prone to over-confident predictions on out-of-distribution and random-noise data. We conclude that the typical overconfidence seen in neural networks, even on in-distribution data is likely a consequence of training with hard labels, suggesting that mixup be employed for classification tasks where predictive uncertainty is a significant concern.

## 1   Introduction: Overconfidence and Uncertainty in Deep Learning

Machine learning algorithms are replacing or expected to increasingly replace humans in decision-making pipelines. With the deployment of AI-based systems in high risk fields such as medical diagnosis [26], autonomous vehicle control [21] and the legal sector [1], the major challenges of the upcoming era are thus going to be in issues of uncertainty and trust-worthiness of a classifier. With deep neural networks having established supremacy in many pattern recognition tasks, it is the predictive uncertainty of these types of classifiers that will be of increasing importance. The DNN must not only be accurate, but also indicate when it is likely to get the wrong answer. This allows the decision-making to be routed as needed to a human or another more accurate, but possibly more expensive, classifier, with the assumption being that the additional cost incurred is greatly surpassed by the consequences of a wrong prediction.

For this reason, quantifying the predictive uncertainty for deep neural networks has seen increased attention in recent years [6, 19, 7, 20, 16, 31]. One of the first works to examine the issue of *calibration* for modern neural networks was [9]; noting that in a well-calibrated classifier, predictive

---

[*]Correspondence to sunil@lanl.gov

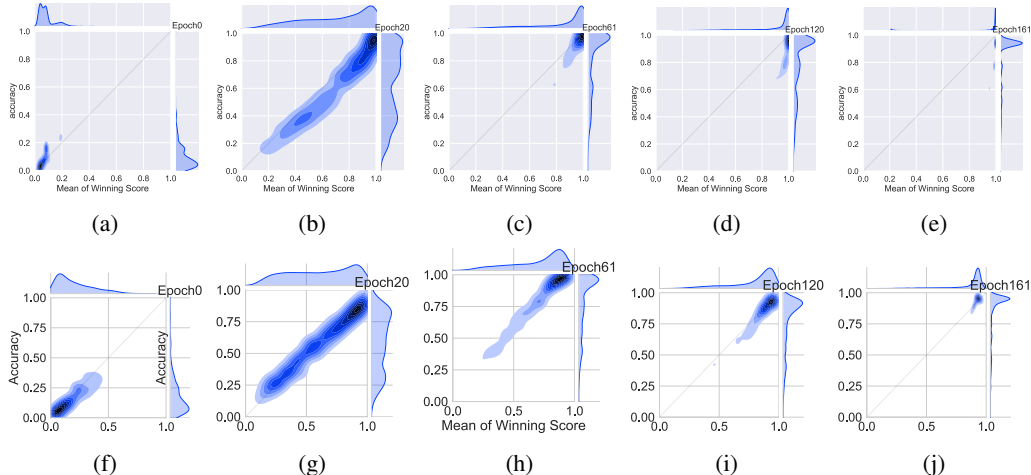

(a)             (b)             (c)             (d)             (e)

(f)             (g)             (h)             (i)             (j)

Figure 1: Joint density plots of accuracy vs confidence (captured by the mean of the winning softmax score) on the CIFAR-100 validation set at different training epochs for the VGG-16 deep neural network. **Top Row:** In regular training, the DNN moves from under-confidence, at the beginning of training, to overconfidence at the end. A well-calibrated classifier would have most of the density lying on the $x = y$ gray line. **Bottom Row:** Training with mixup on the same architecture and dataset. At corresponding epochs, the network is much better calibrated.

scores should be indicative of the actual likelihood of correctness, the authors in [9] show significant empirical evidence that modern deep neural networks are poorly calibrated, with depth, weight decay and batch normalization all influencing calibration. Modern architectures, it turns out, are prone to overconfidence, meaning accuracy is likely to be lower than what is indicated by the predictive score. The top row in Figure 1 illustrates this phenomena: shown are a series of joint density plots of the average winning score and accuracy of a VGG-16 [32] network over the CIFAR-100 [18] validation set, plotted at different epochs. Both the confidence (captured by the winning score) as well as accuracy start out low and gradually increase as the network learns. However, what is interesting – and concerning – is that the confidence always leads accuracy in the later stages of training; accuracy saturates while confidence continues to increase resulting in a very sharply peaked distribution of winning scores and an overconfident model.

While tempering overconfidence in neural networks using alternatives to the final softmax layer has been studied before [25], here we investigate the effect of entropy of the training labels on calibration. Most modern DNNs, when trained for classification in a supervised learning setting, are trained using one-hot encoded labels that have all the probability mass in one class; the training labels are thus zero-entropy signals that admit no uncertainty about the input. The DNN is thus, in some sense, trained to become overconfident. Hence a worthwhile line of exploration is whether principled approaches to label smoothing can somehow temper overconfidence. Label smoothing and related work has been explored before [33, 30]. In this work, we carry out an exploration along these lines by investigating the effect of the recently proposed *mixup* [40] method of training deep neural networks. In mixup, additional synthetic samples are generated during training by convexly combining random pairs of images and, importantly, their labels as well. While simple to implement, it has shown to be a surprisingly effective method of data augmentation: DNNs trained with mixup show noticeable gains in classification performance on a number of image classification benchmarks. However neither the original work nor any subsequent extensions to mixup [36, 10, 23] have explored the effect of mixup on predictive uncertainty and DNN calibration; this is precisely what we address in this paper.

Our findings are as follows: mixup trained DNNs are significantly better calibrated – i.e the predicted softmax scores are much better indicators of the actual likelihood of a correct prediction – than DNNs trained without mixup (see Figure 1 bottom row for an example). We also observe that merely mixing features does not result in the same calibration benefit and that the label smoothing in mixup training plays a significant role in improving calibration. Further, we also observe that mixup-trained DNNs are less prone to over-confident predictions on out-of-distribution and random-noise data. We note

here that in this work we do not consider the calibration and uncertainty over adversarially perturbed inputs; we leave that for future exploration.

The rest of the paper is organized as follows: Section 2 provides a brief overview of the mixup training process; Section 3 discusses calibration metrics, experimental setup and mixup's calibration benefits for image data with additional results on natural language data described in Section 4; in Section 5, we explore in more detail the effect of mixup-based label smoothing on calibration, and further discuss the effect of training time on calibration in Section 6; in Section 7 we show additional evidence for the benefit of mixup training on predictive uncertainty when dealing with out-of-distribution data. Further discussions and conclusions are in Section 8.

## 2 An Overview of Mixup Training

Mixup training [40] is based on the principle of Vicinal Risk Minimization [3](VRM): the classifier is trained not only on the training data, but also in the *vicinity* of each training sample. The vicinal points are generated according to the following simple rule introduced in [40]:

$$\tilde{x} = \lambda x_i + (1 - \lambda)x_j$$
$$\tilde{y} = \lambda y_i + (1 - \lambda)y_j$$

where $x_i$ and $x_j$ are two randomly sampled input points, and $y_i$ and $y_j$ are their associated one-hot encoded labels. This has the effect of the empirical Dirac delta distribution

$$P_\delta(x, y) = \frac{1}{n} \sum_i^n \delta(x = x_i, y = y_i)$$

centered at $(x_i, y_i)$ being replaced with the *empirical vicinal distribution*

$$P_\nu(\tilde{x}, \tilde{y}) = \frac{1}{n} \sum_i^n \nu(\tilde{x}, \tilde{y}|x_i, y_i)$$

where $\nu$ is a *vicinity distribution* that gives the probability of finding the virtual feature-target pair $(\tilde{x}, \tilde{y})$ in the vicinity of the original pair $(x_i, y_i)$. The vicinal samples $(\tilde{x}, \tilde{y})$ are generated as above, and during training minimization is performed on the *empirical vicinal risk* using the vicinal dataset $\mathcal{D}_\nu := \{(\tilde{x}_i, \tilde{y}_i)\}_{i=1}^m$:

$$R_\nu(f) = \frac{1}{m} \sum_{i=1}^m \mathcal{L}(f(\tilde{x}_i), \tilde{y}_i)$$

where $L$ is the standard cross-entropy loss, but calculated on the soft-labels $\tilde{y}_i$ instead of hard labels. Training this way not only augments the feature set $\tilde{X}$, but the induced set of soft-labels also encourages the strength of the classification regions to vary linearly betweens samples. The experiments in [40] and related work in [15, 36, 10] show noticeable performance gains in various image classification tasks. The linear interpolator $\lambda \in [0, 1]$ that determines the mixing ratio is drawn from a symmetric Beta distribution, $Beta(\alpha, \alpha)$ at each training iteration, where $\alpha$ is the hyper-parameter that controls the strength of the interpolation between pairs of images and the associated smoothing of the training labels. $\alpha = 0$ recovers the base case corresponding to zero-entropy training labels (one-hot encodings, in which case the resulting image is either just $x_i$ or $x_j$), while a high value of $\alpha$ ends up in always averaging the inputs and labels. The authors in [40] remark that relatively smaller values of $\alpha \in [0.1, 0.4]$ gave the best performing results for classification, while high values of $\alpha$ resulted in significant under-fitting. In this work, we also look at the effect of $\alpha$ on calibration performance.

## 3 Experiments

We perform numerous experiments to analyze the effect of mixup training on the calibration of the resulting trained classifiers on both image and natural language data. We experiment with various deep architectures and standard datasets, including large-scale training with ImageNet. In all the experiments in this paper, we only apply mixup to *pairs* of images as done in [40]. The mixup functionality was implemented using the mixup authors' code available at [39].

## 3.1 Setup

For the small-scale image experiments, we use the following datasets in our experiments: STL-10 [4], CIFAR-10 and CIFAR-100 [18] and Fashion-MNIST [37]. For STL-10, we use the VGG-16 [32] network. CIFAR-10 and CIFAR-100 experiments were carried out on VGG-16 as well as ResNet-34 [12] models. For Fashion-MNIST, we used a ResNet-18 [12] model. For all experiments, we use batch normalization, weight decay of $5 \times 10^{-4}$ and trained the network using SGD with Nesterov momentum, training for 200 epochs with an initial learning rate of 0.1 halved at 2 at 60,120 and 160 epochs. Unless otherwise noted, calibration results are reported for the best performing epoch on the validation set.

## 3.2 Calibration Metrics

We measure the calibration of the network as follows (and as described in [9]): predictions are grouped into $M$ interval bins of equal size. Let $B_m$ be the set of samples whose prediction scores (the winning softmax score) fall into bin $m$. The accuracy and confidence of $B_m$ are defined as

$$\mathrm{acc}(B_m) = \frac{1}{|B_m|} \sum_{i \in B_m} \mathbf{1}(\hat{y}_i = y_i)$$

$$\mathrm{conf}(B_m) = \frac{1}{|B_m|} \sum_{i \in B_m} \hat{p}_i$$

where $\hat{p}_i$ is the confidence (winning score) of sample $i$. The **Expected Calibration Error** (ECE) is then defined as:

$$\mathrm{ECE} = \sum_{m=1}^{M} \frac{|B_m|}{n} \left| \mathrm{acc}(B_m) - \mathrm{conf}(B_m) \right|$$

In high-risk applications, confident but wrong predictions can be especially harmful; thus we also define an additional calibration metric – the **Overconfidence Error** (OE) – as follows

$$\mathrm{OE} = \sum_{m=1}^{M} \frac{|B_m|}{n} \left[ \mathrm{conf}(B_m) \times \max \Big( \mathrm{conf}(B_m) - \mathrm{acc}(B_m), 0 \Big) \right]$$

This penalizes predictions by the weight of the confidence but only when confidence exceeds accuracy; thus overconfident bins incur a high penalty.

## 3.3 Comparison Methods

Since mixup produces smoothed labels over mixtures of inputs, we compare the calibration performance of mixup to two other label smoothing techniques:

- $\epsilon-$*label smoothing* described in [33], where the one-hot encoded training signal is smoothed by distributing an $\epsilon$ mass over the other (i.e., non ground-truth) classes, and

- *entropy-regularized loss (ERL)* described in [30] that discourages the neural network from being over-confident by penalizing low-entropy distributions.

Our baseline comparison (no mixup) is regular training where no label smoothing or mixing of features is applied. We also note that in this section we do not compare against the temperature scaling method described in [9], which is a post-training calibration method and will generally produce well-calibrated scores. Here we would like to see the effect of label smoothing while training; experiments with temperature scaling are reported in Section 7.

## 3.4 Results

Results on the various datasets and architectures are shown in Figure 2. While the performance gains in validation accuracy are generally consistent with the results reported in [40], here we focus on the effect of mixup on network calibration. The top row shows a calibration scatter plot for STL-10 and CIFAR-100, highlighting the effect of mixup training. In a well calibrated model, where the confidence matches the accuracy most of the points will be on $x = y$ line. We see that in the base

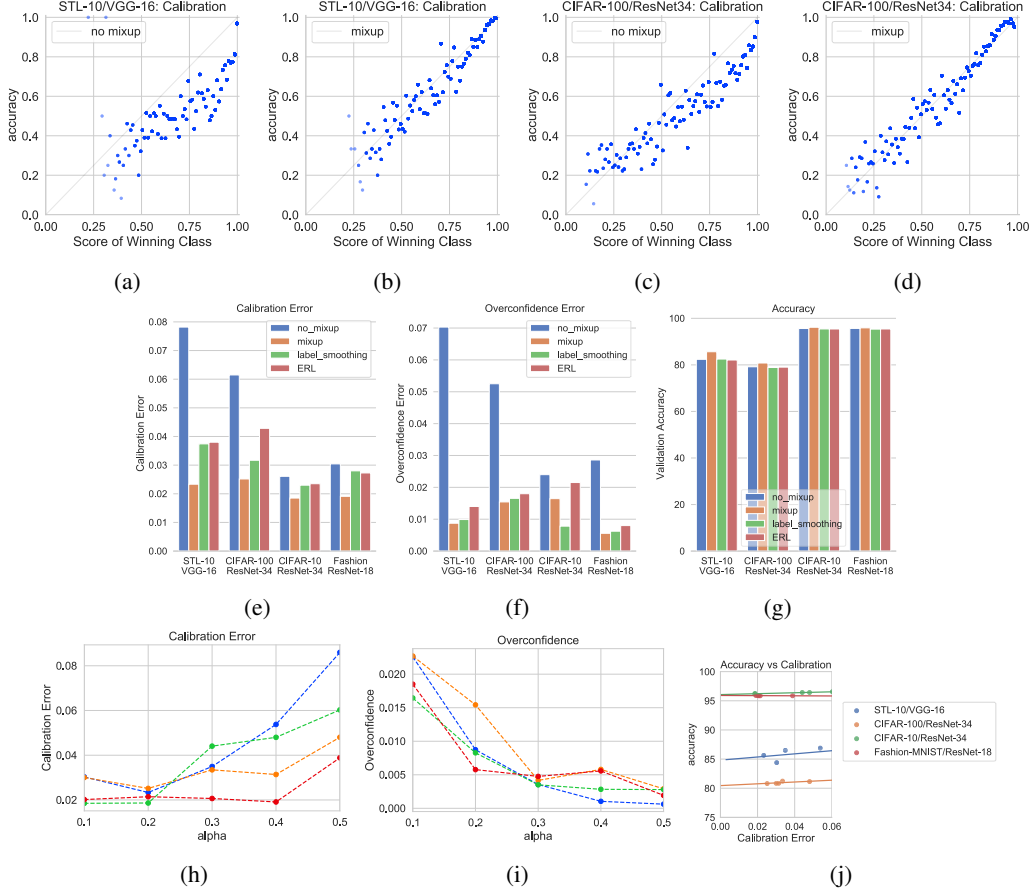

Figure 2: Calibration results for mixup and baseline (no mixup) on various image datasets and architectures. **Top Row**: Scatterplots for accuracy and confidence for STL-10(a,b) and CIFAR-100(c,d). The mixup case is much better calibrated with the points lying closer to the $x = y$ line, while in the baseline, points tend to lie in the overconfident region. **Middle Row:** Mixup versus comparison methods where label_smoothing is the $\epsilon$-label smoothing method and ERL is the entropy regularized loss. **Bottom Row**: Expected calibration error (e) and overconfidence error (f) on various architectures. Experiments suggest best ECE is achieved for $\alpha$ in the [0.2,0.4] (h), while overconfidence error decreases monotonically with $\alpha$ due to under-fitting (i). Accuracy behavior for differently calibrated models is shown in (j).

case, both for STL-10 and CIFAR-100, most of the points tend to lie in the overconfident region. The mixup case is much better calibrated, noticeably in the high-confidence regions. The bar plots in the middle row provide results for accuracy and calibration for various combinations of datasets and architectures against comparison methods. We report the calibration error for the best performing model (in terms of validation accuracy). For label smoothing, an $\epsilon \in [0.05, 0.1]$ performed best while for ERL, the best-performing confidence penalty hyper-parameter was $0.1$. The trends in the comparison are clear: label smoothing either via $\epsilon$-smoothing, ERL or mixup generally provides a calibration advantage and tempers overconfidence, with the latter generally performing the best in comparison to other methods. We also show the effect on ECE as we vary the hyperparameter $\alpha$ of the mixing parameter distribution. For very low values of $\alpha$, the behavior is similar to the base case (as expected), but ECE also noticeably worsens for higher values of $\alpha$ due to the model being *under-confident*. Indeed, mixup models can be under-confident if $\alpha$ is large which is related to manifold intrusion [10]: for large $\alpha$, a mixed-up sample is more likely to lie away from the original manifold and thus be affected by manifold intrusion, where a mixed sample collides with a real sample on the data manifold, but is given a soft label that is different from the label of the real example. Overconfidence alone decreases monotonically as we increase $\alpha$ as shown in Figure 2i. We also show the accuracy of mixup models at various levels of calibration determined by $\alpha$. As can be

seen, a well-tuned $\alpha$ can result in a better-calibrated model with very little loss in performance. Our classification results here are consistent with those reported in [40] where the best performing $\alpha$ was in the $[0.1, .0.4]$ range.

### 3.4.1 Large-scale Experiments on ImageNet

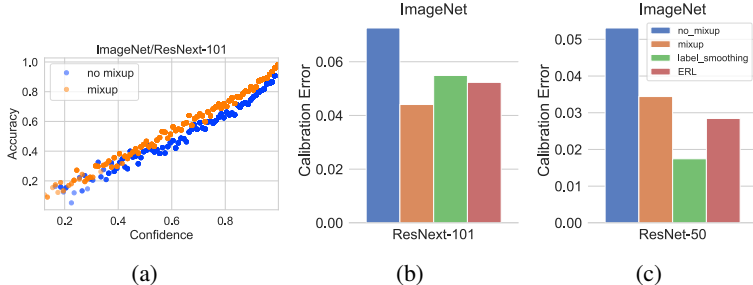

Figure 3: Calibration on ImageNet for ResNet architectures

Here we report the results of calibration metrics resulting from mixup training on the 1000-class version of the ImageNet [5] data comprising of over 1.2 million images. One of the advantages of mixup and its implementation is that it adds very little overhead to the training time, and thus can be easily applied to large scale datasets like ImageNet. We perform distributed parallel training using the synchronous version of stochastic gradient descent. We use the learning-rate schedule described in [8] on a 32-GPU cluster and train till 93% accuracy is reached over the top-5 predictions. We test on two modern state-of-the-art architectures: ResNet-50 [12] and ResNext-101 (32x4d) [38]. The results are shown in Figure 3. The scatter-plot showing calibration for ResNext-101 architecture suggests that mixup training provides noticeable benefits even in the large-data scenario, where the models should be less prone to over-fitting the one-hot labels. On the deeper ResNext-101, mixup provides better calibration than the label smoothing models, though this same effect was not visible for the ResNet-50 model. However, both calibration error and overconfidence show noticeable improvements using label smoothing over the baseline. The mixup model did however achieve a consistently higher classification performance of $\approx 0.4$ percent over the other methods.

## 4 Experiments on Natural Language Data

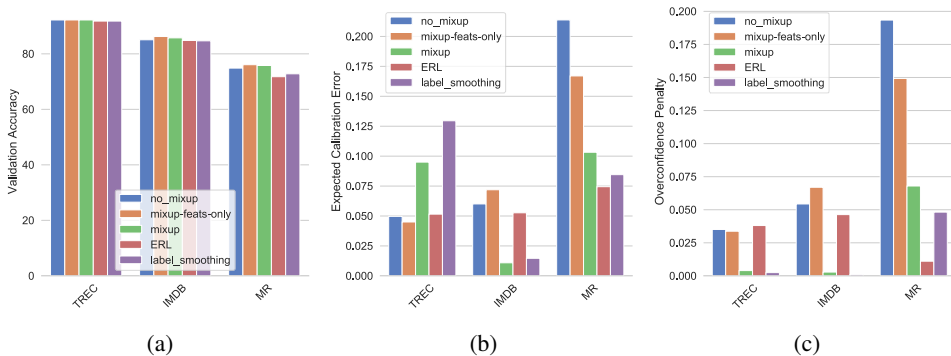

Figure 4: Accuracy, calibration and overconfidence on various NLP datasets

While mixup was originally suggested as a method to mostly improve performance on image classification tasks, here we explore the effect of mixup training in the natural language processing (NLP) domain. A straight-forward mixing of inputs (as in pixel-mixing in images) will generally produce nonsense input since the semantics are unclear. To avoid this, we modify the mixup strategy to perform mixup on the embeddings layer rather than directly on the input documents. We note that this approach is similar to the recent work described in [11] that utilizes mixup for improving

sentence classification which is among the few works, besides ours, studying the effects of mixup in the NLP domain. For our experiments, we employ mixup on NLP data for text classification using the MR [28], TREC [22] and IMDB [24] datasets. We train a CNN for sentence classification (Sentence-level CNN) [17], where we initialize all the words with pre-trained GloVe [29] embeddings, which are modified while training on each dataset. For the remaining parameters, we use the values suggested in [17]. We refrain from training the most recent NLP models [14, 2, 41] since our aim here is not to show state-of-art classification performance on these datasets, but to study the effect on calibration. We show these results in Figure 4 where it is evident that mixup provides noticeable gains for all datasets, both in terms of calibration and overconfidence. We leave further exploration of principled strategies for mixup for NLP as future work.

## 5 Effect of Soft Labels on Calibration

So far we have seen that mixup consistently leads to better calibrated networks compared to the base case, in addition to improving classification performance as has been observed in a number of works [36, 10, 23]. This behavior is not surprising given that mixup is a form of data augmentation: in mixup training, due to random sampling of both images as well as the mixing parameter $\lambda$, the probability that the learner sees the same image twice is small. This has a strong regularizing effect in terms of preventing memorization and over-fitting, even for high-capacity neural networks. Indeed, unlike regular training, the training loss in the mixup case is always significantly higher than the base case as observed by the mixup authors [40]. Because of the significant amount of data augmentation resulting from the random combination in mixup, from the perspective of statistical learning theory, the improved calibration of a mixup classifier can be viewed as the classifier learning the true posteriors $P(Y|X)$ in the infinite data limit [35]. However this leads to the following question: if the improved calibration is essentially an effect of data augmentation, does simply combining the images without combining the labels provide the same calibration benefit?

We perform a series of experiments on various image datasets and architectures to explore this question. Results from the earlier sections show that existing label smoothing techniques that increase the entropy of the training signal do provide better calibration without exploiting any data augmentation effects and thus we expect to see this effect in the mixup case as well. In the latter case, the entropies of the training labels are determined by the $\alpha$ parameter of the $Beta(\alpha, \alpha)$ distribution from which the mixing parameter is sampled. The distribution of training entropies for a few cases of $\alpha$ are shown in Figure 5. The base-case is equivalent to $\alpha = 0$ (not shown) where the entropy distribution is a point-mass at 0.

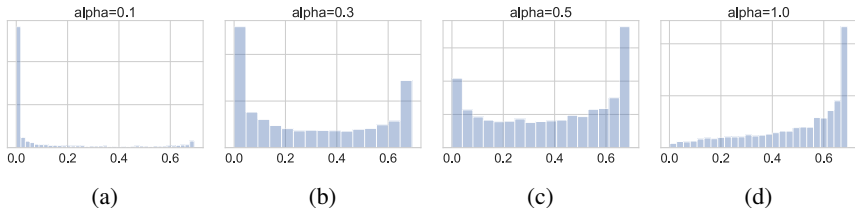

Figure 5: Entropy distribution of training labels as a function of the $\alpha$ parameter of the $Beta(\alpha, \alpha)$ distribution from which the mixing parameter is sampled.

To tease out the effect of full mixup versus only mixing features, we convexly combine images as before, but the resulting image assumes the hard label of the nearer class; this provides data augmentation without the label smoothing effect. Results on a number of benchmarks and architectures are shown in Figure 6. The results are clear: merely mixing features does not provide the calibration benefit seen in the full-mixup case suggesting that the point-mass distributions in hard-coded labels are contributing factors to overconfidence. As in label smoothing and entropy regularization, having (or enforcing via a loss penalty) a non-zero mass in more than one class prevents the largest pre-softmax logit from becoming much larger than the others tempering overconfidence and leading to improved calibration.

In addition to feature and label mixing, a recent extension to mixup [36] also proposes convexly combining the representations in the hidden layer of the network; we report the calibration effects of this approach in the supplementary material.

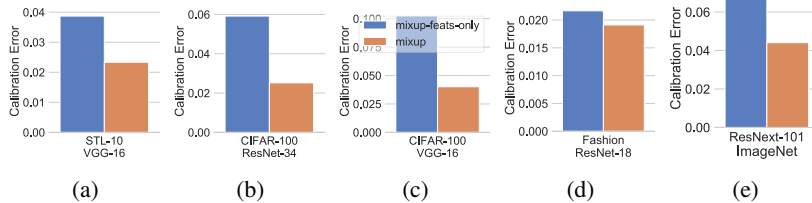

(a)      (b)      (c)      (d)      (e)

Figure 6: Calibration performance when only features are mixed vs. full mixup, on various datasets and architectures

# 6 Effect of Extended Training on Mixup Calibration

As remarked in the previous section, one of the contributing factors to improved calibration in mixup is the significant data augmentation aspect of mixup training, where the model is unlikely to see the same mixed-up sample more than once. The natural question here is whether these models will eventually become overconfident if trained for much longer periods. In Figure 7, we show the training curves for a few extended training experiments where the models were trained for 1000 epochs: for the baseline (i.e when $\alpha = 0$.), the train loss and accuracy approach 0 and 100% respectively (i.e., over-fitting), while in the mixup case (non-zero $\alpha$'s), the strong data augmentation prevents over-fitting. This behavior is sustained over the entire duration of the training as can be seen in the corresponding values of ECE. Mixup models, even when trained for much longer, continue to have a low calibration error, suggesting that the mixing of data has a sustained inhibitive effect on over-fitting the training data (the training loss for mixup continues to be significantly higher than baseline even after extended training) and preventing the model from becoming overconfident.

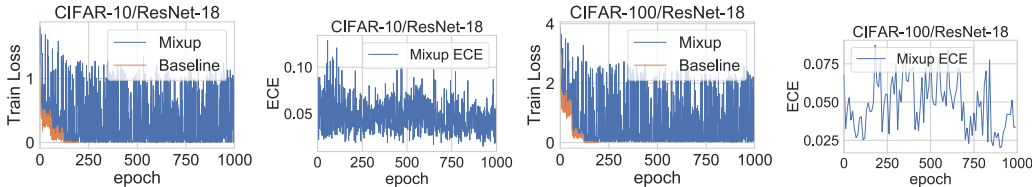

Figure 7: Training loss and calibration error under extended training for CIFAR-10 and CIFAR-100 with mixup. Baseline (no mixup) training loss (orange) goes to zero early on while mixup continues to have non-zero training loss even after 1000 epochs. Meanwhile, calibration error for mixup does not exhibit an upward trend even after extended training.

# 7 Testing on Out-of-Distribution and Random Data

In this section, we explore the effect of mixup training when predicting on samples from unseen classes (out-of-distribution) and random noise images. Deep networks have been shown to produce pathologically overconfident predictions on random noise images [13], and here we would like to explore the effect of mixup training on such behavior. We first train a VGG-16 network on in-distribution data (STL-10) and then predict on classes sampled from the ImageNet database that have not been encountered

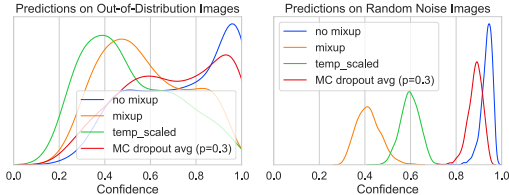

Figure 8: Distribution of winning scores from various models when tested on out-of-distribution and gaussian noise samples, after being trained on the STL-10 dataset.

during training. For the random noise images, we test on gaussian random noise with the same mean and variance as the training set.

We compare the performance of a mixup-trained model with that of the baseline, as well as a temperature calibrated pre-trained baseline as described in [9]. Since the latter is a post-training calibration method, we expect it to be well-calibrated on in-distribution data. We also compare the prediction uncertainty using the Monte Carlo dropout method described in [6] where multiple forward passes using dropout are made during test-time. We average predictions over 10 runs. The distribution over prediction scores for out-of-distribution and random data for mixup and comparison methods are shown in Figure 8. The differences versus the baseline are striking; in both cases, the mixup DNN is noticeably less confident than its non-mixup counterpart, with the score distribution being nearly perfectly separable in the random noise case. While temperature scaling is more conservative than mixup on real but out-of-sample data, it is noticeably more overconfident in the random-noise case. Further, mixup performs significantly better than MC-dropout in both cases.

In Table 1, we also show a comparison of the performance of the aforementioned models for reliably detecting out-of-distributon and random-noise data, using Area under the ROC (AUROC) curve as the metric. Mixup is the best performing model in both cases, significantly outperforming the others as a random-noise detector. Temperature scaling, while producing well-calibrated models for in-distribution data is not a reliable detector. The scaling process reduces the confidence on both in and out-of-distribution data, significantly reducing the ability to discriminate between these two types of data. Mixup, on

| Method | AUROC(In/Out) | |
|---|---|---|
| | STL-10/ ImageNet | STL-10/ Gaussian |
| Baseline | 80.57 | 73.28 |
| Mixup ($\alpha$=0.4) | **83.28** | **95.93** |
| Temp. Scaling | 56.2 | 54.2 |
| Dropout($p$=0.3) | 78.93 | 70.57 |

Table 1: Out-of-category detection results for the DAC on STL-10 and Tiny ImageNet.

the other hand, does well in both cases. The results here suggest that the effect of training with interpolated samples and the resulting label smoothing tempers over-confidence in regions away from the training data. While these experiments were limited to two datasets and one architecture, the results indicate that training by minimizing vicinal risk can be an effective way to enhance reliability of predictions in DNNs. Note that since mixup trains the model by convexly combining pairs of images, the synthesized images all lie within the convex hull of the training data. In the supplementary material, we provide results on the prediction confidence when images lie outside the convex hull of the training set.

## 8    Conclusion and Future Work

We presented results on an unexplored area of mixup based training: its effect on DNN calibration and predictive uncertainty. Existing empirical work has conclusively shown the benefits of mixup for boosting classification performance; in this work, we show an additional important benefit – mixup trained networks are better calibrated and provide more reliable estimates both for in-sample and out-of-sample data (being under-confident in the latter case). There are possibly multiple reasons for this: the data augmentation provided by mixup is a form of regularization that prevents over-fitting and memorization, tempering overconfidence in the process. The label smoothing resulting from mixup might be viewed as a form of entropic regularization on the training signals, again preventing the DNN from driving the training error to zero. The results in this paper provide further evidence that training with hard labels is likely one of the contributing factors leading to overconfidence seen in modern neural networks. Recent work [36] has shown how the classification regions in mixup are smoother, without sudden jumps from one high confidence region to another suggesting that the lack of sharp transition boundaries in classification regions play an important role in producing well-calibrated classifiers. Recent works such as [27] also confirm the benefit of label-smoothing on calibration.

Since mixup is implemented while training, it can also be employed with post-training calibration like temperature scaling, model perturbations like the dropout method or even the ensemble models described in [19]. Further, mixup-based models can also be combined with rejection classifiers, both during training – such as the abstention approached proposed in [34] for dealing with label noise – as well as during inference [7]. Indeed, the boost in classification performance coupled with the well-calibrated nature of mixup-trained DNNs as studied in this paper suggests that mixup is a highly effective approach for training deep neural networks where predictive uncertainty is a significant concern.

**Acknowledgments**

We would like to thank the anonymous referees for their valuable suggestions for improving the paper. The authors were supported in part by the Joint Design of Advanced Computing Solutions for Cancer (JDACS4C) program established by the U.S. Department of Energy (DOE) and the National Cancer Institute (NCI) of the National Institutes of Health. This work was performed under the auspices of the U.S. Department of Energy by Los Alamos National Laboratory under Contract DE-AC5206NA25396. This work was also supported in part by the CONIX Research Center, one of six centers in JUMP, a Semiconductor Research Corporation (SRC) program sponsored by DARPA.

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
