[Supplementary Material]

## A   Additional Experiments on Mixup Calibration

For completeness, we provide results using CIFAR-10 and CIFAR-100 on the ResNet-18 architecture as used in existing literature on mixup [38, 34, 10], although these works did not consider the calibration aspect.

| Method | Test Accuracy | ECE |
| --- | --- | --- |
| No Mixup (Baseline) | 95.12 | 0.023 |
| Mixup ($\alpha$=0.4) | 96.16 | 0.019 |
| Mixup ($\alpha$=1.0) | 96.04 | 0.1 |
| Label Smoothing ($\epsilon$=0.1) | 95.51 | 0.089 |
| ERL ($\kappa$=0.1) | 95.55 | 0.046 |

(a) CIFAR-10/ResNet-18

| Method | Test Accuracy | ECE |
| --- | --- | --- |
| Baseline | 78.28 | 0.049 |
| Mixup (alpha=0.5) | 79.57 | 0.035 |
| Mixup (alpha=1.0) | 79.54 | 0.091 |
| Label Smoothing (eps=0.1) | 79.08 | 0.066 |
| ERL (kappa=1.0) | 78.47 | 0.6 |

(b) CIFAR-100/ResNet-18

Table 2: Mixup results on CIFAR datasets with the ResNet 18 architecture

Results are shown in Table 2. The baseline performance (no mixup) matches the baseline accuracies reported in the previous literature. We also provide the expected calibration error (ECE) for the best performing model as well as the mixup model that used $\alpha = 1.0$ that was used in the previous literature. We find that lower $\alpha$ gives slightly better classification and signficantly better ECE. Note that ECE can be high due to both the model being overconfident as well as under-confident, the latter being the case for $\alpha = 1.0$ since this causes the resulting training signal to have higher entropies than with smaller $\alpha$'s.

## B   Prediction Confidence of Mixup

Figure 9: Distribution of winning scores on various image datasets

As we have seen, mixup trained models are less overconfident than their non-mixup counterparts. Here we show the distribution of the winning scores for various image datasets. As shown in Figure 9, mixup models are less peaked in the very-high confidence region.

## C   Mixing in the Hidden Layers: Manifold Mixup and Effects on Calibration

The empirical results in this paper show that convexly combining features and labels significantly improves model calibration and predictive uncertainty of deep neural networks since the higher entropy training signal makes the model less confident in the regions of interpolated data. One natural extension to the basic mixup idea is *manifold mixup* proposed in [34], where the representations in the *hidden layers* are also combined linearly. The authors demonstrate that interpolation in hidden

| Dataset | Method | Accuracy | ECE |
|---|---|---|---|
| CIFAR-10 (PreActResNet-18) | Mixup (200 epochs) | 96.16 | 0.02 |
| | Manifold Mixup (200 epochs) | 95.96 | 0.047 |
| | Manifold Mixup (2000 epochs) | 97.07 | 0.077 |
| CIFAR-10 (PreActResNet-34) | Mixup (200 epochs) | 96.42 | 0.04 |
| | Manifold Mixup (200 epochs) | 96.2 | 0.04 |
| | Manifold Mixup (2000 epochs) | 97.5 | 0.007 |
| CIFAR-100 (PreActResNet-18) | Mixup (200 epochs) | 79.57 | 0.047 |
| | Manifold Mixup (200 epochs) | 74.79 | 0.22 |
| | Manifold Mixup (2000 epochs) | 79.73 | 0.06 |
| CIFAR-100 (PreActResNet-34) | Mixup (200 epochs) | 81.22 | 0.03 |
| | Manifold Mixup (200 epochs) | 77.36 | 0.187 |
| | Manifold Mixup (2000 epochs) | 82.32 | 0.03 |

Table 3: Manifold mixup experiments. For the extended training experiments, we use the same setup as [34] while for the experiments that trained for 200 epochs, we anneal the learning rate at epoch 60, 120 and 160 while keeping all other hyperparameters fixed to match the regular mixup experiments in Section 3

layers smooths the decision boundaries and encourages the model to learn class representations with fewer directions of variance. Here we emprically investigate the effect of this additional training signal from the hidden layers on model calibration. We use the PreActResNet architectures with the CIFAR-10 and CIFAR-100 datasets identical to those used in [34]. We train the models for both 200 epochs using the same learning rate as we used in our earlier experiments, as well as for 2000 epochs using the learning rate schedule in [34] and report on calibration and accuracy in both cases.

Results are in Table 3. When trained for the same number of epochs as the regular mixup experiments reported in this paper, manifold mixup generally has lower accuracy and worse calibration errors. However, accuracy is significantly better after training for 2000 epochs (as done in [34]), with ECE improving in a few cases. However, this is not a consistent trend, and further, since the manifold mixup algorithm is more complicated, involves more hyperparameters and takes longer to train than regular mixup, in practice, regular mixup might provide a more practical approach for improved calibration.

# D  Leaving the Convex Hull

Figure 10: Prediction behavior as one moves away from the training data

Since mixup trains the model by convexly combining pairs of images, the synthesized images all lie within the convex hull of the training data. In this section, we explore the behavior of mixup as we gradually leave the convex hull in a random direction.

Specifically, given an input image $\mathbf{X} \in \mathbb{R}^m$, we choose a random vector $\mathbf{d} \in \mathbb{R}^m$ (where $d_i \sim U(-1, 1)$), and perturb $\mathbf{X}$ as follows: $\mathbf{X}' = \mathbf{X} + \mu \hat{\mathbf{d}}$. We try this for different $\mathbf{d}$'s and $\mu$'s and observe the predictions for a pre-trained mixup model and explore how the prediction behavior changes.

We test three versions of a pre-trained VGG-16 model: mixup, baseline (no mixup) and a temperature-scaled version of the baseline, all trained on STL-10 data. We experiment over a wide range of the perturbation parameter $\mu$. Figure 10 shows how the prediction accuracies, winning softmax scores (confidence) and the entropy of the prediction distributions change in all three cases.

As the images are more perturbed (and thus become more noisy), the accuracy of mixup is more robust and does not degrade as quickly as the other methods. Note that the baseline and temperature-scaled versions will have identical predictions and thus identical accuracies, since temperature scaling does not change the winning class, but only scales the softmax scores. This is evident in the confidence plot where temperature scaling quickly loses confidence as the perturbations get larger. Mixup confidence decays more gradually, similar to its decay in accuracy. The base model loses confidence, and then quickly regains it as the images get further away from the training set – a pathological behavior of deep neural networks that has been widely observed in the literature. Threshold-based confidence models will obviously fail in such cases. Prediction entropy shows similar behavior to confidence. It is worthwhile to note that even a small perturbation of 0.01 (where the image structure is largely preserved) quickly degrades the confidence of temperature scaled models, indicating that they are less robust to additive noise; a threshold-based prediction mechanism will reject a significant number of samples in such cases. At a large perturbation value (100), the accuracy of mixup is still about 25% while the base model (and thus the temperature scaled versions) are no better than random (10%)