[Reviews · NeurIPS 2019]

Reviewer 1



Mixup is originally proposed as a data augmentation / regularization method. In the past few years, it has been widely adopted as an effective method to improve generalization performance. While the mixup method is not new, this work seems to be the first to use it for training well-calibrated models and convincingly demonstrate its applicability and advantage over several alternatives. Specifically, the paper shows on image classification as well as NLP datasets that mixup training significantly improves expected calibration error (ECE) and overconfidence error (OE) on the validation set; that mixup training also leads to much more conservative uncertainty estimations on out-of-distribution inputs; and that the improvement is likely due to training with soft-labels. Given that accurate uncertainty estimation is an important topic for risk-sensitive machine learning applications, this work may lead to significant practical impact. Overall I find the paper is written with good clarity, the experimental results strong and the impact could be high, hence I vote for acceptance.

Reviewer 2



The paper studies a newly proposed data augmentation method Mixup from a calibration perspective. The authors empirically show that models trained with Mixup are significantly better calibrated, in terms of the correlation between the predicted softmax score and the actual likelihood of a correct prediction, when compared to models trained without Mixup. The empirical studies of the paper also show that mixup-trained models are less prone to over-confident prediction on out-of-distribution data, and the label smoothing in Mixup is critical for achieving the desirable calibration outcomes. The paper is well written and easy to follow. The new view of looking into the newly proposed data augmentation method Mixup is interesting and could be practically useful for application where uncertainty is a significant concern. Nevertheless, I have the following concerns on the paper. 1. In terms of predictive accuracy, Mixup performing well for out-of-manifold data, providing effective regularization effect, and heavily benefiting from training on mixed (soft) labels to obtain good performance have been well studied and observed by its original paper [28] and its follow-up variants such as [24] and [9]. The authors here just provide observations using a different evaluation metric, namely the model calibration. In this sense, I think the technical novel is a bit limited. The paper would be much significant if it provides some insights on why and how Mixup could achieve these calibration effects, either theoretically or empirically. 2. I have concerns on the baselines used and the experimental setups. I am not so sure if the networks for computing the calibration metrics are well trained. The trained models used are the ones with the best performance on the validation set. I wonder if the selected models are underfitting/overfitting, which could have significant impact on the calibration effect. In this sense, it would be useful to also provide the training/loss curves of the training process. 3. Another problem with the baselines used here is that, the networks architectures used are different from the ones used by the Mixup paper [28] and its improved variants such as Manifold Mixup[24] and AdaMixup[9], where ResNet-18 is used for both the Cifar10 and Cifar100, and the ResNet-18 networks are well trained/tuned by careful parameter turnings such as learning rate reschedules. 4. It would be useful to show when the baseline model starts to generate over-confident predictions. Is that correlated with the training/loss curves? Is it a result of overfitting? For example, in Figure 1, training with 160 epochs may not be a problem for Mixup due to the newly generated mixed samples in each new epoch, but it may overfit the baseline model easily. Providing the training curves for Figure 1 and Figure 2 could be useful. 5. The choice of Alpha between [0.1 04.] in the paper is different from the original Mixup paper[28]. For both the Cifar10 and Cifar100, the original paper used Alpha=1 (not as claimed in this paper at the end of section3.4). The choice of the mixing policy generation parameter Alpha is very sensitive to Mixup as shown in the paper. This observation suggests that it would be useful to provide experimental results for Alpha larger than 0.5 in the bottom figures of Figure 2. Presentation issue: 1. It could be helpful to move/merge Section 3.2 into the Introduction Section since the introduction section discusses the numbers computed with the calibration metrics presented in Section 3.2. Question to the authors: 1. Would Mixup generate over-confident predictions if the models are trained long enough? *********************after rebuttal******************************* I would like to thank you for the rebuttal. The authors’ responses have addressed most of my concerns on the empirical study setups, targeting the items listed in the Improvement Section of the review form. Therefore, I am willing to increase my score. In general, the observations in this paper are quite novel, although I still have concern on its technical novelty. I also like the experimental studies on NLP tasks, although Mixup’s effects, in terms of predictive accuracy, on NLP classification tasks have been reported in a recent paper: https://arxiv.org/abs/1905.08941 *********************************************************************

Reviewer 3



Post rebuttal: I read the rebuttal and other reviewers' comments. Assuming that the authors will fix the paper formatting and move the NLP experiments to the main body of the manuscript -- I'm satisfied with the paper and I'll stay with my original recommendation. ------------------------------- The paper discusses the calibration and predictive uncertainty aspects of the mixup training. Those are two important aspects of NNs, especially when deployed in decision making systems. The paper is well written and the insights are well validated. The results suggest that the models trained with mixup achieve better uncertainty calibration and are less prone to overconfidence. Strengths: + The paper paper is well written and easy to follow. + The idea of studying the uncertainty calibration for mixup training is very reasonable and shows good results. Weaknesses: - The formatting of the paper seems to be off. - The paper could benefit from reorganization of the experimental section; e. g. introducing NLP experiments in the main body of the paper. - Since the paper is proposing a new interpretation of mixup training, it could benefit by extending the comparisons in Figure 2 by including [8] and model ensembles (e. g. [14]). Some suggestions on how the paper could be improved: * Paper formatting seems to be off - It does not follow the NeurIPS formatting style. The abstract font is too large and the bottom page margins seem to be altered. By fixing the paper style the authors should gain some space and the NLP experiments could be included in the main body of the paper. * It would be interesting to see plots similar to 2(h) and 2(i) for ERL and label smoothing. * Figure 2 - it would be beneficial to include comparisons to [8] and [14] in the plots. * Figure 3 - it would be beneficial to include comparisons to [8] and [14]. * Adding calibration plots (similar to Figure 2 (a:d) ) would make the paper stronger. * Section 3.4.1 - What is the intuition behind the change of trends when comparing the results in Figure 3(b) to 3(c)? * Adding calibration results for manifold mixup [24] would further improve the paper. * Figure 5: Caption is missing. The mixup-feats-only results could be included in the Figure 2 -- this would lead to additional space that would enable moving some content from the supplementary material to the main body. * Figure 2(j) is too small. * l101 - a dot is missing * l142 - The bar plots in the bottom row -> The bar plots in the middle row

[Author Response · NeurIPS 2019]

We thank the reviewers for their comments and actionable suggestions on improving the paper. Below we address the most pressing concerns. We paraphrase some of the comments for brevity.

| Method | Test Accuracy | ECE |
|---|---|---|
| Baseline | 95.12 | 0.023 |
| Mixup ($\alpha$=0.4) | 96.16 | 0.019 |
| Mixup ($\alpha$=1.0) | 96.04 | 0.1 |
| Label Smoothing ($\epsilon$=0.1) | 95.51 | 0.089 |
| ERL (kappa=0.1) | 95.55 | 0.046 |

(a) CIFAR-10/ResNet-18

| Method | Test Accuracy | ECE |
|---|---|---|
| Baseline | 78.28 | 0.049 |
| Mixup (alpha=0.5) | 79.57 | 0.035 |
| Mixup (alpha=1.0) | 79.54 | 0.091 |
| Label Smoothing (eps=0.1) | 79.08 | 0.066 |
| ERL (kappa=1.0) | 78.47 | 0.6 |

(b) CIFAR-100/ResNet-18

**Comment**: "Justify that the baseline models are well trained, and compare with existing baselines that use ResNet-18 for CIFAR-10/100" (R2). We provide additional results on ResNet-18 for both CIFAR-10 and 100. Our baselines (w. no mixup) match the baseline accuracies reported in related work. We also provide the expected calibration error (ECE) for the best performing model as well as the mixup model that used $\alpha = 1.0$ as suggested by reviewer 2. We find that lower $\alpha$ gives slightly better classification and signficantly better ECE. Note that ECE can be high both due to the model being overconfident as well as under-confident, the latter being the case for $\alpha = 1.0$ since this causes the resulting training signal to have higher entropies than with smaller $\alpha$.

Figure 1: Train loss and accuracy for mixup for various alphas. Baseline corresponds to $\alpha = 0$

**Comment**: "Provide training loss curves for Figures 1 and 2. " (R2)We show training curves for some of the experiments in the paper in above Figure. Your intuition is correct: for the baseline (i.e when $\alpha = 0$.), over-fitting on the training set is indeed correlated with transitioning to overconfidence. The baseline train loss and accuracy approach 0 and 100% respectively (i.e., over-fitting), while in the mixup case (non-zero $\alpha$'s), the strong data augmentation prevents over-fitting and thus restricts the model from making overconfident predictions. This behavior is sustained even if one trains for much longer (see next section)

**Comment**: "Will the mixup models become overconfident if trained for longer?" (R2) Below we provide the ECE vs epoch for both CIFAR-10 and CIFAR-100 for the mixup models trained for 1000 epochs (original experiments only used 200 epochs). We see that the mixup model, even when trained for much longer, continues to have a low calibration error, suggesting that the mixing of data has a sustained inhibitive effect on over-fitting the training data (the training loss for mixup continues to be significantly higher than baseline even after extended training) and preventing the model from becoming overconfident.

Figure 2

As for why mixup improves calibration (R2), please see discussion in Section 4:the strong data augmentation and label smoothening are both contributive factors: one can view mixup training as training with infinite data (since the model never sees the same data point twice) in which case true posteriors are learnt according to statistical learning theory, but in addition the label softening (which prevents the winning logits from becoming arbitrarily large) also prevents overconfidence. Note that mixup models can turn out to be *underconfident* if $\alpha$ is large. In fact, this is also related to manifold intrusion: a mixed-up sample is more likely to lie away from the original manifold and thus be affected by manifold intrusion if $\alpha$ is large. In our experiments, we see the resulting models are prone to under-fitting and under-confidence. We will include a discussion on ROC and AUC curves for mixup in the final version (R1). As for comparing calibration of mixup with temperature scaling (R3), this produces almost perfectly calibrated scores since it is a post-training calibration approach. We will incorporate comparisons with model ensembles and manifold mixup (R3) in the final version; we expect the latter to also produce well-calibrated scores since it is a generalization of mixup.

[Meta-Review · NeurIPS 2019]

This paper investigates the use of mixup to improve the calibration of neural nets. Neural nets are known to be poorly calibrated and this poses significant problems in several important applications. The reviewers found that this work provides compelling empirical evidence that mixup address this important problem. The concerns raised by the reviewers were sufficiently addressed by the rebuttal. This work would be of interest to the NeurIPS community.